# Mouse Models of Mayaro Virus

**DOI:** 10.3390/v15091803

**Published:** 2023-08-24

**Authors:** Rafael Borges Rosa, Emilene Ferreira de Castro, Débora de Oliveira Santos, Murilo Vieira da Silva, Lindomar José Pena

**Affiliations:** 1Department of Virology and Experimental Therapy (LAVITE), Aggeu Magalhães Institute (IAM), Oswaldo Cruz Foundation (Fiocruz), Recife 50740-465, Brazil; 2Rodents Animal Facilities Complex, Federal University of Uberlandia (REBIR-UFU), Uberlandia 38400-902, Brazil; murilo.vieira@ufu.br; 3Faculty of Medicine, Federal University of Uberlandia, Uberlandia 38400-902, Brazil; emilene.castro@ufu.br; 4Dental Hospital, Oral Pathology Laboratory, University of Uberlandia, Uberlandia 38400-902, Brazil; debora.oliveira.santos@ufu.br

**Keywords:** Mayaro, public health, animal model, drugs and vaccines

## Abstract

Mayaro virus (MAYV), the etiologic agent of Mayaro fever, leads patients to severe myalgia and arthralgia, which can have a major impact on public health in all the countries where the virus circulates. The emergence and dissemination of new viruses have led the scientific community to develop new in vivo models that can help in the fight against new diseases. So far, mice have been the most used animal model in studies with MAYV and have proved to be an adequate model for recapitulating several aspects of the disease observed in humans. Mice are widely used in in vivo research and, therefore, are well known in the scientific community, which has allowed for different strains to be investigated in the study of MAYV. In this review, we summarize the main studies with MAYV using mice as an experimental model and discuss how they can contribute to the advancement of the understanding of its pathogenesis and the development of new drugs and vaccines.

## 1. Introduction

MAYV is the etiological agent of Mayaro fever, an acute febrile infectious disease. Mayaro fever is a mild, self-limited febrile syndrome and has two phases. The acute phase presents a short transient viremia [1,2]. After an incubation period of up to 14 days, there is a sudden onset of fever between 39 and 40.2 °C (102.2 and 104.36 °F) [3], concomitant with frontal headache, arthralgia, myalgia, joint swelling, chills, retroorbital pain, malaise, rash, vomiting, and diarrhea [1,4,5]. In some cases, there is the presence of nausea, cough, sore throat, abdominal pain, nasal congestion, appearance, anorexia, swollen lymph nodes, and bleeding gums [1,3,5]. Approximately 20% of cases have swelling of the joints, mainly in the wrists, fingers, ankles, and toes [3]. After the acute phase, a convalescence phase begins, which may be accompanied by arthralgia and arthritis, lasting for several weeks or months [2,3,6]. These symptoms can be similar to those of other arboviral diseases, including but not limited to chikungunya, dengue, and Zika.

MAYV is part of the antigenic complex called the Semliki Forest, along with the chikungunya, Sindbis, and Ross River viruses [7]. First isolated in Trinidad and Tobago in 1954, MAYV has been reported in several countries in the tropical regions of South and Central America. Human infections are thought to occur mostly through the enzootic cycle. Few epidemiological data are available due to inadequate surveillance and the generic nature of its clinical manifestations, which result in misdiagnoses with other viral fevers [8]. In Brazil, MAYV was isolated the following year, in 1955, on the banks of the Guama River in the city of Belem. The Mayaro virus was the cause of an outbreak of acute febrile illness that affected 100 rural workers in the region. Since then, some sporadic cases, outbreaks, and small epidemics of Mayaro fever have been reported in several countries in the tropical regions of South and Central America. Clinical cases of the febrile illness and virus isolation have already been reported in countries such as Brazil, Peru, Suriname, French Guiana, Guyana, Venezuela, Colombia, Ecuador, Panama, and Bolivia. Serological investigations have also indicated the presence of the virus in Costa Rica, Guatemala, and Mexico [8,9].

MAYV can infect, replicate, and spread in both vertebrate and invertebrate hosts. In endemic regions of Latin America, its transmission cycle occurs mainly in wild and rural areas. In this type of cycle, the hematophagous mosquito *Haemagogus janthinomys* is classified as the main vector and non-human primates are considered to be the primary hosts [10]. However, to a lesser extent, other types of mosquitoes may participate as occasional vectors, such as those of the genera *Mansonia* and *Culex*. Some vertebrates can be secondary hosts (viral reservoir), such as rodents, reptiles, and birds. Infection in humans is classified as accidental and occurs when humans are exposed to wild reservoir habitats [11,12].

Phylogenetic studies of MAYV based on the E2/E1 gene sequence have validated the existence of three genotypes: D (widely dispersed), N (new), and L (limited). Genotype D is found in different countries in South America. The N genotype has been isolated only in Peru. The L genotype is more restricted to the northern region of Brazil, where it was isolated for the first time, specifically in the state of Pará in 1955 [13].

There are indications that MAYV is spreading and generating new strains. Currently, MAYV is considered as emerging and there are several factors that indicate the possibility of the urbanization of this virus that can configure the disease as a major public health problem and a possible cause of epidemics [14]. It should be considered that the Mayaro virus is part of the same Semliki Forest Complex as the chikungunya virus, which is known to have an established history of adapting to the urban environment. In addition, studies have shown the possibility of the transmission of MAYV via *Aedes aegypti* [15] and occurrences of cases of infection with MAYV have already been reported in regions close to large cities infested by this same vector [13]. The possibility of the urban adaptation of the vector, together with anthropogenic factors such as rapid urbanization, climate change, and increased population mobility, increase the risk of the distribution of the virus to other continents, generating the possibility of disease outbreaks in countries with no history of infection.

Appropriate animal models can develop the viral infection similar to how it happens in humans, considering the pathogenesis of the disease and the clinical signs presented by patients. Using animal models, it is possible to elucidate the biology of the virus and the mechanisms of its infection. Although each animal species has some limitations of use, mice have been widely used and have already contributed much of the knowledge generated about urban arboviruses of public health interest. Such findings are important for the development of studies aimed at finding new preventive and therapeutic therapies capable of preventing and combating the disease caused by arboviruses [16,17]. In this review, we summarize the main in vivo studies already developed using murine models to study pathogenesis, immunity, treatments, vaccines, and the host–vector relationship.

## 2. Mouse Models and Their Applicability

Mice were the animals used in the first isolation of MAYV. Neonates inoculated with the virus showed a high pathogenic degree, but this pathogenicity was not observed for adult mice, even when the inoculation occurred intracerebrally [18]. From then on, other viral strains of MAYV were also isolated, using mice as an animal model [19,20]. Currently, this animal model has been explored, improved, and widely used in several areas of research with MAYV (Table 1).

### 2.1. Pathogenesis and Immunity

An infection model using immunocompetent mice was developed using the 6-week-old Balb/c strain. An intraplantar inoculation of MAYV induced persistent hypernociception, transient viral replication in the target organs, the systemic production of inflammatory cytokines and chemokines, and specific humoral responses of IgM and IgG [21]. Fifteen-day-old Balb/c mice also developed disease caused by MAYV. Infection via two distinct routes, below the forelimb and in the posterior foot pad, resulted in homogeneous viral dissemination and the development of acute disease in the animals. Clinical signs such as bristly hair, stooped posture, eye irritation, and a slight change in gait were observed. The presence of myositis, arthritis, tenosynovitis, and periostitis was also evaluated. The immune response was characterized by a strong inflammatory response mediated by the cytokines TNF-α, IL-6, and INF-γ and the chemokine MCP-1, followed by the action of the cytokines IL-10 and IL-4 [22].

Another pathogenesis study demonstrated that MAYV replicated and induced clinical signs in wild-type mice less than 11 days old (129Sv/Ev strains (A129WT and C57BL6), both immunocompetent strains. These animals demonstrated the ability to restrict MAYV replication as they aged. Furthermore, the study demonstrated that, in addition to the age of the mice, the type I interferon response was related to the restriction of MAYV infection in the infected mice. Mice deficient in the production of T and B lymphocytes (RAG-1^−/−^ strain) and knockout mice for the Ifnar1 allele, lacking type I interferon receptor function, which results in a reduced immune response and an increased susceptibility to viral infection (IFN⍺R1^−/−^ (IFNAR)), were evaluated. According to the results, MAYV persistently replicated in RAG-1 mice, with the virus being detected in blood and tissues up to 40 days after infection, indicating that adaptive immunity is essential for the elimination of MAYV. Despite chronic replication, the infected adult RAG-1 mice did not develop an apparent sign of muscle damage at early and late infection. On the other hand, MAYV infection in IFNAR, 129WT, and C57BL/6 mice triggered an increase in their expressions of pro-inflammatory mediators, such as TNF, IL-6, KC, IL-1β, MCP-1, and RANTES in muscle tissue, and decreases in the expression of TGF-β [24].

Six–eight-week-old C57BL/6 mice also displayed clinical signs of disease following infection. These animals showed evident dorsal plantar edema between the fifth and sixth day after infection. The hypernociception evaluation showed that the mice felt pain after 1 day post-infection, which lasted until the 8th day of evaluation. The viral replication in tissues was measured up to the tenth day of infection. After reaching a peak at 1 day post-infection, there was a drop in the viral titer achieved in the plantar pad and muscles, but the viral load remained high in the spleen throughout the study [27]. Using inflammasome-deficient mice (Nlrp3 ^−/−^ strain), a study conducted in 2019 elucidated that the NLRP3 inflammasome, a critical component of the innate immune system, is involved in swelling, inflammation, and pain in the dorsoplantar region, establishing a role for the NLRP3 inflammasome in the pathogenesis of MAYV. Higher levels of caspase1-p20, IL-1β, and IL18 are detected in the serum of MAYV-infected patients compared to healthy individuals, also demonstrating the participation of the NLRP3 inflammasome during MAYV infection in humans [27].

Additionally, using C57BL/6 mice, one study found that preexisting immunity against CHIKV confers a partial cross-protection against secondary infection caused by MAYV, reducing the disease severity, tissue viral load, and tissue damage. This study also demonstrated that, interestingly, both human and mouse CHIKV antibodies show a low cross-neutralization of MAYV, but this neutralizing activity significantly increased after secondary infection. Furthermore, the depletion of adaptive immune cells (CD4+ T cells, CD8+ T cells, and CD19+ B cells) did not alter the cross-protective phenotype, suggesting that distinct cell subsets or a combination of CHIKV-stimulated adaptive immune cells are responsible for the protection partial against MAYV. The reduction in pro-inflammatory cytokines, such as interferon gamma (IFN-γ), in animals secondarily infected with MAYV also suggests a role for innate immunity in cross-protection [28]. In the study carried out by Webb et al., 2019, a strong cross-protection against MAYV was observed in C57BL/6 mice and immunodeficient mice (IFNα/βR ^−/−^ (A129ABR) and IFNAR strains) pre-exposed to a wild-type CHIKV strain. These data suggest that CHIKV infection may confer cross-protective effects against MAYV, and the resulting reduction in viremia may limit the potential for the emergence of MAYV [29].

Antibody-dependent protection against alphaviruses is thought to be mediated by two main mechanisms: Fab-mediated viral neutralization and Fc-dependent effector functions. Humoral protection against alphaviruses likely reflects contributions from non-neutralizing antibodies via Fc-dependent controls that accelerate viral clearance. Some studies [30,36,37,38] have suggested that Fc neutralization and effector functions can effectively control MAYV infection. DNA vaccines based on MAYV envelope proteins induce robust neutralizing antibodies as well as T-cell responses to multiple epitopes on the MAYV envelope, which confer protection after a challenge. A live attenuated MAYV vaccine based on an IRES approach induced neutralizing antibody levels similar to those in natural infection and protected mice from lethal challenge [32,33]. Adenovirus-vectored MAYV vaccines, based on either a replication-incompetent human adenovirus type 5 vector (Ad5) or chimpanzee adenoviral vector (ChAdOx1) encoding the structural proteins of MAYV, have provided full or partial protection, respectively, against challenges in immunodeficient mice [25,34].

An effective protective response against alphavirus infection can be determined by the location of the antibody binding to the intact and exposed virus proteins on the surface of infected cells, the inherent neutralizing capacity, the mechanism (blocking of binding, entry, fusion, or exit), and likely the accessibility of the Fc region to engage FcγR and mediate effector functions [39]. 

The antibody-dependent enhancement (ADE) of virus infections has been described for several viruses, including influenza, dengue virus (DENV), Zika virus (ZIKV), Ross River virus (RRV), human immunodeficiency virus (HIV), and Marburg virus. Through this mechanism, suboptimal concentrations of neutralizing antibodies (or non-neutralizing antibodies) bind to viruses without blocking or clearing the infection and promote viral entry into cells, increasing the virus replication and disease severity. This phenomenon has not been described for MAYV but has been proven to aggravate CHIKV infection and disease severity in mice models. Thus, preclinical and clinical trials of vaccine candidates against these alphaviruses should be aware of these possible side effects [40]. 

### 2.2. Treatment and Vaccines

Mice have also contributed to research seeking new treatments and vaccines against MAYV. Favipiravir, a broad-spectrum antiviral drug, has been shown to exert anti-MAYV activity in vitro. In vivo, a study using C57BL/6 mice also evaluated the efficacy of favipiravir against MAYV. The results showed significant reductions in the infectious viral particles and viral RNA transcripts in the tissues and blood of pre- and concomitantly treated infected mice. A significant reduction in the presence of the viral RNA transcript and infectious viral particles was also observed in the tissues and blood of previously and concomitantly treated infected mice. On the other hand, treatment with Favipiravir after MAYV infection did not result in reduced viral replication. These results suggest that Favipiravir is a potent antiviral drug when administered in a timely manner [31].

The complex extracted from the Silybum marianum plant, called silymarin, has also been evaluated for its antiviral action. Silymarin helped to improve the infectious condition of Balb/c mice, protecting the animals against liver damage, oxidative stress, inflammation, and viral replication [23].

A live attenuated vaccine (MAYV/IRES) was evaluated, using mice as an experimental model. The inoculation of MAYV/IRES in Balb/c mice induced strong specific cellular and humoral responses. Furthermore, MAYV/IRES vaccination of immunocompetent mice deficient in interferon receptor production (strain A129ABR) resulted in protection against disease induced by the virulent wild-type MAYV strain [21]. Previously vaccinated mice of the immunocompetent strain ICR-CD-1 (28 days old) and immunodeficient A129ABR (5–8 weeks old) produced high titers of neutralizing antibodies. Mice vaccinated and challenged with the wild-type MAYV strain showed complete protection against disease [33].

Vectored vaccines based on chimpanzee adenoviruses that encode the MAYV structural E1, E2, E3, and 6K proteins (ChAdOx1 May) have also been shown to be efficacious. Researchers demonstrated that ChAdOx1 May was able to provide full protection against the MAYV challenge in A129ABR mice. The ChAdOx1 May vaccine also provided partial cross-protection against CHIKV, with this protection being assessed by the following parameters: survival, weight loss, foot edema, and viremia [25]. A vaccine expressing the complete viral structural polyprotein based on a non-replicating human adenovirus V (AdV) platform has also been developed. Vaccination with AdV-MAYV elicited neutralizing antibodies that protected C57BL/6 mice against a challenge with MAYV, preventing viremia, reducing the viral spread to tissues, and mitigating the viral illness. The vaccine also prevented the virus-mediated killing in IFNAR lineage immunodeficient mice. A passive transfer of immune serum from vaccinated animals similarly prevented infection and disease in C57BL/6 mice, as well as virus-induced death in IFNAR mice, indicating that antiviral antibodies are protective. Immunization with AdV-MAYV also generated cross-neutralizing antibodies against two other related arthritogenic alphaviruses, CHIKV and the Una virus. These cross-neutralizing antibodies were protective against lethal infection in IFNAR mice after a challenge with these heterotypic alphaviruses [34].

A DNA vaccine was also evaluated in mice models. The development and testing of a vaccine named scMAYV-E, which encodes a synthetically engineered consensus MAYV envelope sequence, was used to immunize mice. This vaccine induced potent humoral responses, including neutralizing antibodies, as well as robust T cell responses to multiple epitopes on the MAYV envelope. The immune responses induced by scMAYV-E protected susceptible mice (A129ABR strain) against morbidity and mortality following a challenge with MAYV [32].

The action of anti-MAYV monoclonal antibodies was evaluated, using mice as experimental models. Four-week-old C57BL/6 mice were inoculated intraperitoneally with an interferon immunosuppressant (anti-Ifnar1) and the monoclonal antibody MAR5A, one day before a challenge with MAYV that occurred via the foot pad route. Although MAYV is non-lethal in immunocompetent mice, the infected animals treated with the anti-Ifnar1 monoclonal antibodies began to succumb to the disease within 3 days of infection, reaching 100% mortality within 6 days of being infected. For a prophylaxis evaluation, anti-MAYV monoclonal antibodies were inoculated into the animals one day before viral inoculation. Nine out of eighteen mice were protected against the lethal challenge. All the antibodies that protected the mice against lethal MAYV infection were of the IgG2a subclass [30].

### 2.3. Transmission and Vector Competence Studies

The dynamics of infection and transmission between mosquito vectors and mammals have also been extensively studied. IFNAR mice naturally infected with MAYV through the bite of the *Aedes aegypti* showed clinical signs of infection, while all mice exposed to *Culex quinquefasciatus* mosquitoes remained healthy. Clean *Aedes aegypti* mosquitoes were also able to become infected and maintain a 50% infection rate seven days after blood feeding in mice infected with MAYV, demonstrating a complete model of transmission (mosquito–host–mosquito) [26]. Eight-week-old IFNα/β/γR ^−/−^ (AG129) immunodeficient mice (type II interferon knockouts) have also proved to be an efficient model in vector competence studies. Eight days after infection, a 100% infection rate was found in the evaluated animals. Infected *Aedes aegypti* transmitted the virus back to the AG129 mice, completing a full cycle of transmission [35].

## 3. Conclusions and Perspectives

MAYV remains an emerging virus potentially harmful to public health, considering that there is an urbanization of the disease (Mayaro fever) taking place through the vectorization of *Aedes Aegypti*, which are periurban and cosmopolitan mosquitoes. However, great progress has already been observed in the development of murine models capable of mimicking aspects of the pathology caused by MAYV (Figure 1). These models have provided valuable information about viral pathogenesis and immune responses and have helped in the development of new antivirals and vaccines.

There are limitations that must be considered when choosing an animal model during experimental design. The experimental model may vary according to the hypothesis and the goals to be investigated. Knockout mice for the interferon pathway, for example, are highly susceptible to viral infection and can be widely used in viral pathogenesis assays, but the fact that there is a limitation of the immune response must be evaluated with parsimony in vaccine tests, considering the possibility of an underestimated interpretation of the effectiveness of the vaccine, since the deficiency in the production of interferon receptors may directly interfere with the expected immune response after immunization, considering the pre-existing failure in the defense mechanism that is mediated by interferon. On the other hand, these animals display clinical signs of disease and succumb to the disease, making them a clear-cut model for the assessment of vaccines and antivirals.

On the other hand, mice with an intact immune system show less evident clinical signs in a shorter period of disease manifestation. Considering that these animals, like humans, are immunocompetent, there is a need to better explore the characterization of the model. The alternatives so far have been the use of young animals, in which the immune system is still vulnerable, and the use of immunosuppressants in adult animals. An alternative to be considered is the serial passage of the virus in immunocompetent mice, in search of an adapted strain capable of causing the disease in these animals.

In general, mice are widespread models in animal experimentation centers around the world, have a known genome, and are cost-effective to maintain. These animals can be considered as an essential tool in the fight against MAYV, as they have already helped and have been helping in investigations with other arboviruses.

## Figures and Tables

**Figure 1 viruses-15-01803-f001:**
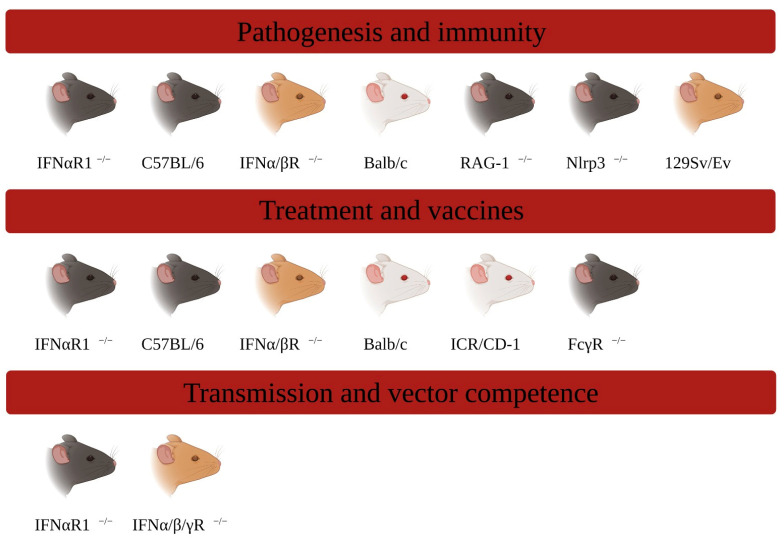
A Mouse models for Mayaro virus. Genetically modified or wild-type, immunocompetent, or immunodeficient strains of different ages have already modeled assays of pathogenesis, immunity, host–vector relationship, and development of new drugs and vaccines. Created with BioRender.com.

**Table 1 viruses-15-01803-t001:** Mouse model of MAYV.

Mouse Strain	Age	Viral Strain	Route of Infection/Dose	Major Findings	Reference
Balb/c	6-week-old	MAYV wt human isolate from Peru 2001	Subcutaneous plantar surface of the right hind paw/50 μL 2 × 10^5^ p.f.u	MAYV induced transient viral replication with persistent hypernociception and production of inflammatory cytokines, chemokines, and humoral responses. An attenuated vaccine induced specific cellular and humoral responses and resulted in protection in immunocompetent mice.	[21]
	15-day-old	TR4675	Subcutaneously, right footpad/10 μL 2.57 × 10^6^ p.f.u of virus suspension in PBS and Subcutaneously in the chest, below the right forelimb/50 μL 1.25 × 10^7^ p.f.u of viral suspension	MAYV infection resulted in the development of acute disease with overt clinical signs. Histopathological studies demonstrated joint impairment mediated by a strong inflammatory response.	[22]
	3-week-old	Acre27	Subcutaneously in the left footpad/20 μL 10^6^ p.f.u	Silymarin helped to improve the condition of MAYV infection in BALB/c mice.	[23]
129 Sv/Ev	6-, 11-, 21-day-old	TR4675	Subcutaneously in the left footpad/20 μL 10^6^ p.f.u	11-day-old immunocompetent mice show clinical signs of MAYV infection, but the phenotypes are completely unapparent after 21 days of age.	[24]
IFNαR1 ^−/−^	8-week-old	TR4675	Subcutaneously in the left footpad/20 μL 10^5^ p.f.u	Infected mice showed a continuous increase in viremia followed by lethality. The animals showed clinical signs of the disease and histopathology showed tissue damage.	[24]
	5-week-old	Mayaro, BeAr505411, NR- 49910	Injection right posterior footpad/20 μL 10^4^ p.f.u	Mice vaccinated with AdV-MAYV were protected from viral infection. Vaccination was able to prevent viral spread and passive transfer of immune serum was able to significantly reduce the pathological symptoms of the infection.	[25]
	2–4 week-old	MAYV/BR/Sinop/H307/2015 (MH513597)	Natural infection (*Aedes aegypti*)	Mice infected with MAYV showed clinical signs of the disease and the virus was transmitted to Aedes aegypti mosquitoes during a blood meal, closing the transmission cycle.	[26]
C57BL/6	8-week-old	TR4675	Subcutaneously in the left footpad/20μL 10^6^ p.f.u	MAYV inoculation in wt C57BL/6 adult mice did not cause any clinical signs of infection.	[24]
	6–8-week-old	BeAr 20290	Subcutaneously ventral side of the footpad/10 μL 10^5^ or 10^6^ p.f.u	Mice infected with MAYV showed swelling and pain in the foot pad. The viral load was transient in all analyzed tissues, falling during the course of the infection, remaining stable only in the spleen.	[27]
	10–12-week-old	TRVL 4675	Ventral side of each foot/20 μL 10^5^ p.f.u	The pre-existing immunity to CHIKV conferred cross-protection against secondary MAYV infection by reducing disease severity, tissue viral load, and histopathological scores in infected mice.	[28]
	4-week-old	12A	Intradermally in the hind footpad/10^4^ p.f.u	Immunity derived from CHIKV infection (WT) decreased disease and prevented viremin in MAYV-infected mice. Immunity induced by highly immunogenic and effective CHIKV vaccines did not provide protection against the virus.	[29]
	4-week-old	MAYV (Beh428890, BeH473130, BeH343155, BeH506151, FSB0311, IQU2950, OBS6443, TRVL15537, BeH407 e Uruma)	Left footpad/10^3^ f.f.u	A panel of anti-MAYV neutralizing mAbs was able to neutralize 11 different strains of MAYV. A subset of mAbs were strongly neutralizing and kept mice protected from lethal challenge.	[30]
	5-week-old	Mayaro, BeAr505411, NR-49910	Injection right posterior footpad/20 μL 10^4^ p.f.u	Robust neutralizing antibody titers were obtained for mice vaccinated with AdV-MAYV. Serum viral titers of vaccinated animals were not detected.	[25]
	4-week-old	UVE/MAYV/1954/TT/TC625	Subcutaneously in the right hind footpad/20 µL 10^6^ p.f.u	The antiviral Favipiravir contributed to the reduction in infection in pre- and concomitantly treated animals. In contrast, post-infection treatment did not result in reduced viral replication.	[31]
RAG-1 ^−/−^	8-week-old	TR4675	Subcutaneously in the left footpad/20 μL 10^6^ p.f.u	MAYV persistently replicated in RAG^−/−^ mice. Despite this, infection in adult mice did not cause lethality and the appearance of clinical signs.	[24]
Nlrp3 ^−/−^	6–8-week-old	BeAr 20,290	Subcutaneously ventral side of the footpad/10 μL 10^5^ or 10^6^ p.f.u	Mice infected with MAYV showed edema in the paws with high presence of neutrophils and tissue damage. Joint washes of the approved high number of inflammatory cells.	[27]
IFNα/βR^−/−^	6-week-old	MAYV wt human isolate from Peru 2001	Subcutaneous plantar surface of the right hind paw/50 μL 2 × 10^5^ p.f.u	The lethality of mice immunized with the MAYV/IRES strain was significantly delayed. Mice infected with MAYV (WT) showed high lethality and high viral load.	[21]
	4-week-old	12A	Intradermally in the hind footpad/10^4^ p.f.u	After challenge with a lethal dose of MAYV, groups vaccinated with EILV/CHIKV showed clinical signs of the disease and consequent mortality. In contrast, MAYV/IRES completely protected against the disease. Passive transfer of CHIKV immune serum was not protective against MAYV challenge in immunocompromised mice.	[29]
	4–6-week-old	Trinidad Regional Virus Laboratory (TRVL) 15,537 MAYV	Intraperitoneal/100 µL 10^2^ p.f.u	The scMAYV-E vaccine protected mice from morbidity and mortality after MAYV challenge. Electroporation-enhanced immunization of mice with this vaccine induced potent humoral responses.	[32]
	5–8-week-old	MAYV wt human isolate from Peru 2001	Intradermally on the left footpad/10^4^ p.f.u	Mice infected with the attenuated MAYV/IRES strain and the wt MAYV strain lost weight. There was no significant difference in footpad swelling. All vaccinated mice survived, whereas all sham mice died on the seventh day after infection.	[33]
	5-week-old	MAYV CH-IQT4235	Intradermally left foot/20 µL 1.6 × 10^4^ p.f.u	The ChAdOx1 May construct provided rapid and robust immunity with high titers of neutralizing antibodies against MAYV, capable of protecting A129 mice from a lethal attack and reducing viremia to undetectable levels. Furthermore, vaccination with ChAdOx1 may offer cross-protection against a lethal CHIKV challenge.	[34]
ICR/CD-1	6- and 28-day-old	MAYV wt human isolate from Peru 2001	Dorsum subcutaneously/10^4^ p.f.u	Six-day-old mice infected with MAYV/IRES survived, unlike animals infected with MAYV WT that died. Adult animals infected with MAYV/IRES steadily gained weight throughout the experiment, while mice infected with MAYV WT lost some weight initially but recovered. A single vaccination proved to be immunogenic in adult CD1 mice and efficacy was demonstrated indirectly through passive transfer of immune mouse serum to infant mice, followed by lethal challenge.	[33]
FcγR ^−/−^	4-week-old	BeH407	Subcutaneous/10^3^ f.f.u	Animals treated with isotype control mAb showed 100% mortality, whereas animals treated with MAY-115 or MAY-134 were only partially protected. MAY-117 also failed to protect mice against MAYV.	[30]
IFNα/β/γR ^−/−^	8–9-week-old	TRVL4645	Intraperitoneal/10^5^ p.f.u	AG129 mice developed transient viremia. The observed viremia was large enough to infect *Aedes aegypti* mosquitoes during a blood meal. Infected mosquitoes transmitted MAYV back to uninfected mice, completing a full cycle of transmission.	[35]

p.f.u: plaque formation unit; f.f.u: focus forming unit; MAYV: Mayaro virus; IRES: internal ribosome entry site; IgM: immunoglobulin M; IgG: immunoglobulin G; WT: wild type; AdV: adenovirus; CHIKV: Chikungunya virus; EILV: Eilat virus; RNA: ribonucleic acid; GFP: green fluorescent protein; ChAdOx1: chimpanzee, adenovirus, and Oxford; scMAYV-E: synthetic DNA envelope vaccine; and mAbs: monoclonal antibodies.

## Data Availability

Not applicable.

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
