# Peer review of "Mouse Models of Mayaro Virus"

_viruses, 2023, doi:10.3390/v15091803_

Round 1

Reviewer 1 Report

A well written review article at a right time.

Every aspects of the MAYV pathogenesis were reviewed nicely and summarized the achievement in the field.

1. A reference demonstrating mosquitoes as the vector should be included. Ref 10-12 do not contain this aspect.

2. Some comments on the potential antibody dependent enhancement would be good in line 163 or in the vaccine section.

Author Response

Reviewer 1

Comments and Suggestions for Authors

A well written review article at a right time.

Response: We appreciate the reviewer´s compliments.

Every aspects of the MAYV pathogenesis were reviewed nicely and summarized the achievement in the field.

Response: We really appreciate you taking the time to express that. It is very encouraging for us.

  1. A reference demonstrating mosquitoes as the vector should be included. Ref 10-12 do not contain this aspect.

Response: All the suggested changes have been made. We have edited the text and added references about the insect vectors for MAYV (lines 56-60)  

  1. Some comments on the potential antibody dependent enhancement would be good in line 163 or in the vaccine section.

Response: All the suggested changes have been made. We have expanded the section about MAYV immunity and discussed about the possible role of antibody dependent enhancement in the context of MAYV infection (lines 166-192).

Reviewer 2 Report

This manuscript describes in detail the mouse models used to study the pathogenesis of Mayaro virus. The data provided are invaluable for any researcher wishing to study this virus in the context of its natural or artificial transmission, to compare the pathogenicity of certain virus strains, or to test antivirals or vaccines. The table summarizing the various models is useful, but at present far too detailed, as much of what appears there is recounted in the text. The "major findings" column should be much more concise. In addition, there are some minor typing errors in the text, as species names are not systematically written in italics (note that Aedes aegypti should be written Aedes aegypti, not Aedes aegypti, and that Aedes aegypti often alternates with Ae. agypti, so try to be consistent in the names). In the Conclusion section, lines 253-254, the part of the sentence: considering the possibility
of disease interpretation of vaccine efficacy is unclear and needs further clarification.

Author Response

Comments and Suggestions for Authors

This manuscript describes in detail the mouse models used to study the pathogenesis of Mayaro virus. The data provided are invaluable for any researcher wishing to study this virus in the context of its natural or artificial transmission, to compare the pathogenicity of certain virus strains, or to test antivirals or vaccines. The table summarizing the various models is useful, but at present far too detailed, as much of what appears there is recounted in the text. The "major findings" column should be much more concise. In addition, there are some minor typing errors in the text, as species names are not systematically written in italics (note that Aedes aegypti should be written Aedes aegypti, not Aedes aegypti, and that Aedes aegypti often alternates with Ae. agypti, so try to be consistent in the names). In the Conclusion section, lines 253-254, the part of the sentence: considering the possibility of disease interpretation of vaccine efficacy is unclear and needs further clarification.

 Response: We thank the reviewer for the positive feedback and suggestions. We have edited the table 1, making the “major findings" column more concise. We have thoroughly revised the text and also added a paragraph in the conclusion section regarding interpretation of vaccine efficacy tests using these models  (lines 277-287).